# Estrogenic and Non-Estrogenic Disruptor Effect of Zearalenone on Male Reproduction: A Review

**DOI:** 10.3390/ijms24021578

**Published:** 2023-01-13

**Authors:** András Balló, Kinga Busznyákné Székvári, Péter Czétány, László Márk, Attila Török, Árpád Szántó, Gábor Máté

**Affiliations:** 1Pannon Reproduction Institute, 8300 Tapolca, Hungary; 2Urology Clinic, Clinical Centre, Medical School, University of Pécs, 7621 Pécs, Hungary; 3National Laboratory on Human Reproduction, University of Pécs, 7624 Pécs, Hungary; 4Department of Analytical Biochemistry, Institute of Biochemistry and Medical Chemistry, Medical School, University of Pécs, 7624 Pécs, Hungary; 5MTA-PTE Human Reproduction Scientific Research Group, 7624 Pécs, Hungary

**Keywords:** epigenetic, estrogen, infertility, oxidative stress, reactive oxygen species, sperm, zearalenone

## Abstract

According to some estimates, at least 70% of feedstuffs and finished feeds are contaminated with one or more mycotoxins and, due to its significant prevalence, both animals and humans are highly likely to be exposed to these toxins. In addition to health risks, they also cause economic issues. From a healthcare point of view, zearalenone (ZEA) and its derivatives have been shown to exert many negative effects. Specifically, ZEA has hepatotoxicity, immunotoxicity, genotoxicity, carcinogenicity, intestinal toxicity, reproductive toxicity and endocrine disruption effects. Of these effects, male reproductive deterioration and processes that lead to this have been reviewed in this study. Papers are reviewed that demonstrate estrogenic effects of ZEA due to its analogy to estradiol and how these effects may influence male reproductive cells such as spermatozoa, Sertoli cells and Leydig cells. Data that employ epigenetic effects of ZEA are also discussed. We discuss literature data demonstrating that reactive oxygen species formation in ZEA-exposed cells plays a crucial role in diminished spermatogenesis; reduced sperm motility, viability and mitochondrial membrane potential; altered intracellular antioxidant enzyme activities; and increased rates of apoptosis and DNA fragmentation; thereby resulting in reduced pregnancy.

## 1. Introduction

In this fast-paced modern world, it is becoming extremely difficult to avoid endocrine-disrupting chemicals containing plastics, radiation, environmental toxicants, phytoestrogens, etc. In parallel with this, over the last few decades, declining sperm parameters, earlier puberty of girls, increasing rate of oocyte aneuploidy, genital malformations and decreasing fertility have been seen worldwide. While the data may not be completely understood, it is clear that there is some connection between our way of life and the decrease in reproductive abilities. However, the vast majority of works discussing this problem focus on women, even though successful conception depends 50% on good-quality sperm.

Mycotoxins are becoming increasingly important among environmental toxicants. Contrary to the increasing infertility, the population of Earth is growing rapidly, which also increases the demand for food, and satisfying this need is only possible with more intensive agriculture. Together with climate change, this provides more opportunity for mycotoxin contamination and animal/human exposure. Animals and humans may come into contact with many mycotoxins, but zearalenone (ZEA) is one of the most interesting due to its mode of action. Here we provide an overview of the pathogenesis of zearalenone and its adverse effects on male reproduction.

Studies that met all the following criteria were reviewed: (i) investigated the impact of ZEA on the reproductive system of male model organisms (rodents, pigs, etc.); (ii) investigated the effects of ZEA on male germ cells such as spermatozoa, Sertoli cells and Leydig cells. Studies conforming to any of the following criteria were also included: (iii) investigated the oxidative stress and apoptosis-inducing ability of ZEA on estrogen receptor-expressing and non-expressing cells and cell lines (spermatozoa, yeasts, HepG2, HL-60, Vero, Caco-2, etc.).

## 2. Occurrence of Zearalenone in Food and Feed

Mycotoxins are small-molecule, natural substances that are secondary metabolic products of filamentous fungi. These metabolites are toxic chemical compounds causing disease or death in vertebrates (human), invertebrates, microorganisms, and plants [1]. Currently, more than 300 mycotoxins are known, but of these, scientific interest is mostly focused on those that are carcinogenic and/or highly toxic. These toxins cause millions of dollars in damage worldwide each year through medical and veterinary costs and unsalvageable agricultural crops [2]. Mycotoxins occur everywhere in the world. A quarter of the world’s crops are contaminated with mycotoxins. Their presence must be considered in almost every phase of food production, processing, storage and distribution. Among the molds, there are those that contaminate plants with their toxins already in the field (arable field molds), while others produce toxins during the improper storage of harvested cereals with high humidity (>20%) (warehouse molds). The former group includes the *Fusarium* species, for which zearalenone is an important toxin from an animal and human health point of view [3].

ZEA is mainly produced by *Fusarium* species (*F. graminearum*, *F. culmorum*, *F. cerealis*, *F. equiseti*, *F. crookwellense*, *F. semitectum*, *F. verticillioides*, *F. sporotrichioides*, *F. oxysporum* and *F. acuminatum*). Its average presence in cereals and other raw materials is 15–75% but in corn samples this value can be even higher [4]. The distribution of ZEA in animal feed shows a geographical pattern; the *Fusarium* species have adapted to the temperate climate that occurs primarily in Europe, Asia and North America [5]. ZEA occurs most often in corn and other cereals (e.g., barley, oats, sorghum, wheat, rice), as well as in feed, food (e.g., flour, pasta, bread, cereal) and drinks (beer) made from them, but ZEA might be also be delivered through water contaminated with *Fusarium* fungus [4]. In addition, the metabolites of ZEA also appear in the milk of cows consuming feed contaminated with ZEA, increasing the chance of possible human exposure [6,7,8]. Poultry, especially chickens, were shown to be quite resistant to ZEA. However, the toxin may appear in eggs [9]. It further aggravates the situation that, after milling ZEA-containing cereals, the mycotoxin is concentrated in the fiber-rich parts [10]. During wet milling of maize, the enrichment of mycotoxins in the husk is observed, while after dry milling, ZEA is concentrated in the germ and bran fractions [9,11].

Due to its lactone ring and the double bond at C_11_ and C_12_, ZEA has two isomers: cis and trans. After oral exposure, a biotransformation process takes place and reduced derivatives (α-zearalenol, β-zearalenol, zearalanone, α-zearalanol, β-zearalanol; Figure 1), sulfate and glucuronide conjugates are formed by hepatic processes [12,13]. The ratio of these derivatives differs from species to species. In humans, both α- and β-zearalenol can be detected, but β is the dominant. In pigs, cows and ducks, α-zearalenone is prevalent [14,15,16]. The conversion between ZEA and its metabolites mainly occur in the intestinal tract and bacteria of this system have a key role in this conversion; for this reason the toxin appears quickly in the feces [4]. The reduced metabolites have almost the same, or even stronger, toxic or xenoestrogenic effects as the initial molecule. For instance, α-zearalenol is 60 times as potent as ZEA, but this value for β-zearalenol is only 0.2. In addition, the absorption of these metabolites and conjugates is much faster; the uptake of ZEA in pigs is around 80–85% [17]. Filamentous fungi are also capable of creating ZEA-sulfate conjugates via plant biotransformation, forming “masked mycotoxins”. In plant cells, zearalenone-14-O-β-glucosidase has been detected and further metabolic products such as zearalenone-14-glucuronide, α-zearalenol-glucuronide and β-zearalenol-glucuronide [4,17] can be formed after animal consumption. The analytical detection of these “masked” toxins is difficult due to the lack of analytical standards and, unfortunately, they also appear in feed and some foods, so animal and human exposure can be significant. In addition, after oral administration, ZEA can be detected in the serum, gastrointestinal tract, feces, lung, liver, bile, adipose cells, reproductive cells and tissues, milk, kidney cells and urine. In vertebrates, its half-life is around 86 h [4]. ZEA has hepatotoxicity, immunotoxicity, genotoxicity, carcinogenicity, intestinal toxicity, reproductive toxicity and endocrine disruption effects [18].

It is impossible to completely eliminate mycotoxins from food and feed. Amounts that have not yet caused pathological changes are regulated with limit values in order to minimize public health risks. Considering the toxin content and consumption of the most important foods, the average daily intake value was determined by the Food and Agriculture Organization of the United Nations (FAO) and the World Health Organization (WHO) and JECFA (Joint FAO/WHO Expert Committee on Food Additives) at 0.5 μg/kg of body weight (bw) [19]. In addition, there have been several recommendations regarding the maximum allowed ZEA concentration in certain products. The maximum ZEA concentration in cereals and cereal products, with the exception of maize by-products, is 2 mg/kg; for maize by-products, a value of 3 mg/kg has been determined. The ZEA content of compound feed for piglets, gilts, puppies, kittens, dogs and cats for reproduction has been maximized at 0.1 mg/kg, while for adult dogs and cats this has been set at 0.2 mg/kg. For sows and fattening pigs, this value is 0.25 mg/kg and for calves, dairy cattle, sheep and goats, it is 0.5 mg/kg [20]. The ZEA limit for food intended for direct human consumption is 0.1 mg/kg or below [13,21]. According to previous studies, based on the ZEA content of cereals and eating habits, the estimated daily ZEA intake for European and North American adults is on average 0.8–29 ng/kg bw, while for children, it is 6–55 ng/kg bw [22]. However, according to some studies, the average daily ZAE intake is between 50–200 ng/kg body bw for vegetarians and even as much as 570 ng/kg/day for vegan/macrobiotic dieters [23]. The FAO/WHO has set a provisional tolerable daily intake value of 0.5 μg/kg bw [24]. The European Food Safety Authority (EFSA) has maximized the combined tolerable daily intake value of ZEA and its derivatives at 0.25 μg/kg bw [25].

## 3. Chemical Properties

ZEA [6-(10-hydroxy-6-oxo-trans-1-undecenyl) β-resorcylic-acid-lactone] has a lactone structure: a ketone and methyl-substituted 14-membered macrocyclic lactone ring containing a trans-double bond attached to the resorcinol moiety (Figure 1). ZEA is a white, crystalline solid with a molecular weight of 318.364 g/mol. ZEA does not dissolve in water (<0.02 mg/mL), but dissolves well in alkaline solutions and organic solvents (e.g., ether, benzene, acetonitrile, methyl chloride, chloroform, ethanol, acetone, hexane). Its melting point is 164–165 °C and its boiling point is around 377.53 °C. The xLogP3-AA value is 3.6. Due to the phenolic hydroxyl groups its structure, it is an acidic compound (pKa = 7.62). ZEA is not decomposed during storage, milling or extrusion and, despite its large lactone ring, ZEA is heat stable. The mycotoxin is not degraded during heat treatment at 100 °C but raising the temperature above 175 °C results in a more significant decomposition, which starts at an even lower temperature in alkaline (pH 10.0) conditions. ZEA has UV absorption maxima at 236, 274 and 316 nm; in ethanol, it has fluorescence with excitation at 314 nm and emission at 450 nm [13,26,27]. Although ZEA does not have a steroid structure, it is similar to the hormone 17β-estradiol and capable of competitively binding to estrogen receptors (Figure 1). Therefore, ZEA can have both estrogenic and non-estrogenic effects and modes of action.

## 4. Overview of the Pathogenesis of Zearalenone

### 4.1. Estrogenic Effects

Estradiol has a crucial role in male reproductive function. It exerts effects not only on the reproductive tract but on the brain as well. It has been clearly documented that estradiol synthesis is performed by Sertoli cells, Leydig cells, peripherical tissues and developing germ cells of the seminiferous tubule via their aromatase activity. Estradiol produced by germ cells contributes to the optimal hormonal milieu of somniferous tubules during spermatogenesis [28,29]. Leydig cells are known to produce testosterone under the influence of LH hormone. In addition, as mentioned above, they also have significant aromatase activity, so they also participate in estradiol production. The increasing estradiol has an inhibitory effect on the pituitary gland; the effect of LH on the Leydig cells decreases, and testosterone synthesis will therefore also decrease (Figure 2). The significant decline in testosterone level leads to diminished sperm concentration. Altered serum testosterone/estradiol ratio (<10) has been proven to be a good indicator of deteriorated spermatogenesis [30]. In Sertoli cells, mRNA of aromatase was detected at every stage of spermatogenesis, but in the immature testis, Sertoli cells are responsible for most of the estradiol production [29]. Since spermatogenesis is an androgen-dependent process, the decrease in testosterone level caused by elevated estradiol leads to the downregulation of the androgen receptors in Sertoli cells and sperm cells [31]. Perhaps the most important function of Sertoli cells is their role in blood-testis barrier formation. It has been shown by MacCalman et al. [32,33] that estradiol, in combination with FSH, is mandatory for N-cadherin synthesis responsible for tight junction-type cell-to-cell adhesion.

Similar to estradiol, ZEA has been shown to exert negative effects on sperm cell, Sertoli cells, and Leydig cells. These include altered concentration, viability, motility, DNA integrity and apoptotic rate of sperm cells; such altered biological functions and apoptotic rates of Sertoli and Leydig cells have been reported in several model systems [34,35,36,37,38,39,40,41,42,43,44,45,46,47,48]. These processes are also manifested in altered structure of seminiferous tubules and weight of the testis [49,50]. Estradiol can trigger both genomic and non-genomic signaling pathways since it binds to classical intracellular ERs and plasma membrane localized G protein-coupled ERs, receptors which have been observed on sperm cells, Sertoli cells and Leydig cells [51].

Abnormal spermatogenesis can be also caused by epigenetic modifications during gametogenesis. It can regulate transcription and other nuclear processes in gametes, but also have influences on zygote, embryo and postnatal life. These epigenetic patterns (defects) can be transferred into the oocyte during fertilization by the sperm. This affects early embryo development and be the cause of fertilization failure, early embryogenesis abnormality and other complications during pregnancy [52,53]. Epigenetic mechanisms are perceived as one of the essential collaborators of the human adaptive response to environmental toxins and toxic compounds, and the means of passing on these effects to the next generation [54,55]. Epigenetic modifications may include DNA methylation, histone modifications, noncoding RNAs and protamine codes which regulate gene expression without affecting the gene sequence [56]. Of these modifications, DNA methylation is among the most studied mechanism of epigenetic regulation as it serves as an important participant in the activation and repression of gene expression, and these can transmit epigenetic information. DNA methylation involves the transfer of a methyl group from S-adenosyl-l-methionine to a cytosine residue [57]. The process is catalyzed by DNA methyltransferases; in the mammalian genome, this is mostly limited to the cytosine residues in CpG dinucleotides achieved by the transfer of a methyl group from S-adenyl methionine to the fifth carbon of a cytosine residue to form 5-methylcytosine (5mC) [58]. DNA methylation is reported to be one of the most essential mechanisms for the genome reprogramming, and 5mC and 5-hydroxymethylcytosine (5hmC) are two significant DNA methylation markers. Certain 5mC and 5hmC patterns are crucial in spermatogenesis [59,60].

The loss of DNA methylation during spermatogenesis causes infertility, which comes with a significant reduction in the size of the testis, so important factors in the creation of DNA methylation in the paternal germline can be easily identified. The creation of methylation during spermatogenesis has therefore been extensively studied using standard molecular and genetic methods, and the mechanisms that are behind de novo methylation in male germ cells are well described [60]. Genetic studies have shown that the formation of DNA methylation in sperm is primarily due to the activity of DNA (cytosine-5)-methyltransferase 3A (DNMT3A) and its cofactor, DNA (cytosine-5)-methyltransferase 3-like (DNMT3L) [61,62]. Both de novo DNA methylation and demethylation changes the cytosine methylation pattern through dynamic processes, which are necessary for embryonic development. Because of these, differentiated cells gain permanent and special DNA methylation patterns that modify tissue-specific gene expression [63]. For the correct functioning of the mature sperm and early embryo, it is crucial that the de novo DNA methylation and demethylation is regulated accurately [52,53].

Other important epigenetic modification is histone modifications. Histone methylation on lysine residues can also activate or repress gene expression [64,65]. The programmed gene expression is essential for mature spermatozoa formation, or spermatogenesis. There are three significant histone methylation markers: H3K4, H3K9, and H3K27. The H3K4 marker controls regulatory transcriptional complexes in the testis, which regulate the differentiation of male germ cells [66]. The H3K9 heterochromatin marker plays an important role in meiosis, by maintaining the condensed chromatin states during the segregation of chromosomes [67]. The repression of transcription by the H3K27 marker is crucial for the homeostasis and differentiation of somatic-specific genes [68].

In a healthy body, the development and functioning of the endocrine organs/cells, the connection of the receptor and the hormone and, as a result, the optimal response for the body are determined at the genetic level. This order can be disturbed by endocrine disruptors in a number of ways as described in the following scenarios: (i) they can connect to the receptor, but not to the extent or at the time designated by the body’s needs, thus triggering an atypical response or an unplanned reaction; (ii) they can bind to the receptor, but without activating it, thus making it impossible for the physiological hormone to bind and generate the appropriate response; (iii) they can bind to carrier proteins instead of the natural hormone, thus disrupting the pathway of hormone action; (iv) they can bind to the enzymes of hormone synthesis, thus disrupting the production and/or degradation of the physiological hormone; or (v) they can interfere with the production of hormone receptors. Endocrine disruptors can disrupt the functioning of the endocrine system even in doses significantly lower than the physiological hormone but, at the same time, they can enter the body in doses significantly higher than the physiological ones. Since some members (hormones) of the endocrine system also act within the system, a defect in the production or binding of a single hormone can affect the entire system [69].

Spermatogenesis is under complex hormonal control by gonadotropins and androgens, and estrogens are now known to have a unique role in several spermatogenic processes [70]. As the result of the temporal decline in sperm quality and testosterone levels, and increased rates of testicular cancer among adult men, there are more and more studies that concentrate on endocrine-disrupting chemicals (EDCs) that may harm men’s reproductive health [71,72]. Cellular models and animal toxicological studies have shown that EDCs can have harmful consequences on the male reproductive system. EDCs can disrupt the maintenance of homeostasis and the regulation of developmental processes in the body because of their estrogen-like and/or anti-androgenic characteristics by interfering with the production, release, transport, metabolism, binding, or elimination of natural hormones [73]. Environmental EDCs evoke their actions through the ERs. The two mammalian receptors for estrogen (ERα and ERβ) are widely distributed throughout the reproductive tract [74,75]. Humans and animal populations are continuously exposed to EDCs through the consumption of contaminated food, dermal contact and breathing in polluted air and dust. Examples of environmental EDCs suggested to have adverse effects on the reproductive system in animals, including humans, are pesticides (e.g., dichlorodiphenyltrichloroethane, methoxychlor), fungicides (e.g., vinclozolin), a range of xenoestrogens (EDCs with estrogenic activity, such as bisphenol A (BPA) and ZEA), insecticides (e.g., trichlorfon), herbicides (e.g., atrazine), and plastics (e.g., phthalates) [76,77,78,79,80,81,82].

Xenoestrogens (XEs) are substances that mimic endogenous estrogens; thus, they can affect physiological functions in humans or other animals [83]. XEs greatly disturb endogenous estrogens via non-genomic and/or genomic signaling pathways because of their ability to imitate or block responses [84]. XEs are estrogenic or antiestrogenic and can bind to receptors, manipulating the differentiation and modulation of cell proliferation, apoptosis, cytokine production, and cell-cycle progression that should be regulated by endogenous 17β-estradiol. ER is a ligand-inducible transcription factor which plays a vital role in development and neoplasia by regulating genes that are involved in cell proliferation and differentiation. The estrogenic or antiestrogenic effects of chemicals are caused by the interactions between ER and other compounds [85,86]. Two examples of XEs are ZEA and BPA. BPA is a mass-produced chemical used primarily in the production of polycarbonate plastics and epoxy resins. BPA can leach into food from the inner epoxy-resin protective coatings of cans and consumer products such as polycarbonate tableware, food storage containers, water bottles, and baby bottles. As BPA can bind weakly to ER: ESR1 and ESR2, it is likely to be an EDC [87].

There are some studies that show that XEs can have negative effect on male fertility. The impact of *Fusarium* cultures per os on male pigs showed a 30% decrease in testes weight and reduction of fertility related to decline of sperm quality and viability [88]. In vitro exposure at concentrations ranging from 125 to 250 μM also caused ZEA to alter the capability of fertilization in a time- and dose-dependent manner of the boar sperm as a consequence of their negative impact on viability, motility and acrosome reaction [89]. Pang et al. [90] found that exposure to low-dose ZEA caused a decrease in mouse sperm motility, concentration, and hyperactive rate, and an increase in sperm malformation and mortality. Wang et al. [82] found that, after zebrafish were exposed to an estrogenic compound (combined xenoestrogens) mixture, germ cell proliferation, meiosis and apoptosis was enhanced in the testis, thus disrupting spermatogenesis. They confirmed that long-term exposure to environmentally relevant concentrations of xenoestrogens affects spermatogenesis in zebrafish adults. ZEA was found to damage Sertoli cells and potentially induce apoptosis in both mice and rat [91,92,93].

There are some studies that show evidence that adult BPA exposure adversely affects epididymal sperm counts, indicating an underlying link between BPA exposure and sperm production [94,95,96,97,98]. There are also postnatal and neonatal studies about BPA exposure causing a decline in spermatogenesis [99,100,101]. Salian et al. [102] found that even low doses of BPA exposure during a critical window period of fetal development resulted in harmful reproductive outcomes in male offspring. Ullah et al. [103] found that low doses of BPA and its analogs affects spermatogenesis with outcomes on oxidative stress and the male reproductive system of 22-day-old rats. Oxidative stress in the testis was significantly elevated, and the daily sperm production and number of sperm in the epididymis were reduced. This proved that exposure to BPA and its analogs for a chronic duration could lead to structural and functional modifications in testicular tissue and endocrine remodeling in the male rat reproductive system.

Because of the negative effect of XEs on male fertility, researchers have started to turn to epigenetics to determine how exactly XEs work. There are only a few studies that focus on epigenetic modifications in spermatogenesis caused by XEs. Doshi et al. [104] found that exposure of neonatal male rats to 2.4 μg of BPA/day for the first 5 days of postnatal life revealed significant hypermethylation of the ERα promoter to varying extents (from 40% to 60%) and the Β promoter region with varying extent (from 20% to 65%), along with a two-fold increase in DNMT3A and DNMT3B expression at the transcript and protein level. The study showed that the neonatal exposure of rats to BPA caused abnormal DNA methylation in the testis, implying that epigenetic changes mediated by methylation is one of the possible mechanisms of BPA-induced adverse effects on spermatogenesis and fertility.

ZEA is a non-steroidal estrogenic mycotoxin that has strong estrogenic effects due to its competition with 17β-estradiol for binding to cytosolic ERs, It can deregulate estrogen pathways by disturbing sex hormone synthesis due to its structural similarity with steroids which allow ZEA to bind competitively to 3α- and 3β-hydroxycorticosteroid dehydrogenases [105]. Men et al. [81] found that, in mice, prenatal low dose (lower than no-observed effect level) of ZEA exposure impaired mouse spermatogenesis and decreased mouse semen quality, caused a decline in mouse sperm motility, concentration and hyperactive rate, and an increase in sperm malformation and mortality. The study showed that the low doses of ZEA inhibited the spermatogenesis in offspring by disrupting the meiosis process and reducing sperm motility and concentration, thus diminishing the sperm quality. They measured the DNA methylation marker 5hmC, histone methylation markers H3K9 and H3K27, and the ERα. They found that, after prenatal low-dose ZEA exposure, the 5hmC was reduced, H3K9 and H3K27 were increased, and ERα was decreased in the offspring testis. These data indicated that the changes may be in epigenetic pathways (DNA methylation and histone methylation) and the interactions with the ER signaling pathway. Gao et al. [39] also found that, in mice, pubertal low doses of zearalenone disrupted spermatogenesis through ERα-related genetic and epigenetic pathways. Their study showed that spermatozoa motility and concentration decreased significantly in ZEA-exposed groups. In ZEA-treated groups, the sperm acrosome integrity was transformed and sperm abnormality was higher. Mouse bw was decreased and the testis index was diminished. They found that DNA methylation markers 5mC and 5hmC were decreased, the histone methylation marker H3K27 was increased, and ERα was reduced in mouse testis (Figure 3, Table 1).

In addition to the above-mentioned estrogenic effects of ZEA, there is another estrogenic pathway. In serum, most of the estrogen binds to albumin and the sex-hormone-binding globulin [106]. By analogy with the estrogen–albumin/estrogen–sex hormone-binding-globulin binding, it was shown that ZEA can also bind to all of them, greatly facilitating its delivery and entry into the target cells. In addition, by their binding, the free estradiol level in serum was increased [107,108].

### 4.2. Non-Estrogenic Effects

In the literature, the non-estrogen-specific properties of ZEA have been investigated in three cell types: (i) in cells that presumably do not express the receptor of β-estradiol (e.g., African green monkey kidney; Vero cell line) [109]; (ii) in cells that have been proven to not express the receptor (e.g., hepatocellular carcinoma; HepG2 cell line) [110]; and (iii) in frequently used cell types, which do express the receptor, but only in low amounts such that the receptors are proven to not respond to estrogen and do not give a hormone-specific response (e.g., human epithelial colorectal adenocarcinoma cell line; Caco-2) [111].

Kouadio et al. [112] investigated the general toxic effect of ZEA, deoxynivalenol and fumonisin B1 on Caco-2 cells. Among the three toxins, ZEA proved to be the most toxic and reduced cell viability the most. Depending on the concentration, ZEA inhibits cell division in Caco-2 and Vero cell lines. Flow cytometric studies showed that the number of cells in the G0/G1 phase decreases as a result of the toxin, which is compensated for in the G2/M phase. The proportion of cells in this latter phase is significantly increased, indicating that an error has occurred in DNA replication and the cells are unable to enter the phase of mitosis. The decrease in cell viability is proportional to the arrest in the G2/M phase, which triggers the inhibition of DNA and protein synthesis [109]. ZEA exerts its immunotoxic effect in part through inhibition of cell division by inhibiting mitogen-activated lymphocyte division. It also induces macrophage activation and an increase in the production of various interleukins (IL-2, IL-5) [19,113].

ZEA was shown to be genotoxic because it causes DNA fragmentation, chromosome aberrations, and the formation of micronuclei [109]. ZEA is covalently bound to DNA, during which the cleavage of the formed adducts by repair mechanisms can lead to the formation of micronuclei. The toxin can also cause dysfunction of the mitotic spindle [114]. If the DNA repair system becomes overloaded due to too much DNA damage and cannot repair the errors, apoptosis is induced in the cell [109], just as highly accumulated reactive oxygen species (ROS) leads to programmed cell death. At the same time, apoptosis plays an important role in the normal development and homeostasis of organisms. Several morphological and biochemical changes point to this process [115]. In HepG2 cells, ZEA induces apoptosis in a concentration- and time-dependent manner via the mitochondrial pathway [116]. Since the toxin decomposes in the liver, it can be considered one of the main targets of the toxin, where it can cause lesions and carcinomas [109]. The cells show nuclear morphological changes characteristic of apoptosis: DNA condensation and fragmentation, while the permeability of the plasma membrane does not change [116].

It was also shown in human HepG2 cells that the expression of the p53 gene increases after the exposure to ZEA. p53 is a transcription factor can be found exclusively in mammals, which plays a role in turning on and off a large number of genes involved in cell cycle arrest, DNA repair, or the initiation of apoptosis. If repair fails, proapoptotic Bcl2 protein family genes are expressed, which also activate the mitochondrial pathway [117]. In a recent work by Lee et al. [118], ZEA-exposed GC-1 spermatogonia cell line showed a significant increment in the level of key proteins involved in apoptosis, such as cleaved caspase-3 and -8, BAD, BAX, and phosphorylation of p53 and ERK1/2 (Figure 3).

Oxidative stress processes occur when the balance of ROS and the antioxidant processes that keep them under control is upset within the cell, the cause of which can be either the inhibition of the synthesis of individual antioxidants, or the excessive accumulation of ROS, or possibly both [119]. ZEA-induced oxidative stress can attack almost all cellular structures, including membrane lipids, proteins, and DNA [120]. As a result of ZEA treatment, the amount of malondialdehyde (MDA) increases in Caco-2 cells, which is one of the late biomarkers of oxidative stress indicating lipid peroxidation processes. Reactive oxygen radicals and free radicals produced by lipid peroxidation can cause damage to target cells. MDA alters cell membrane structure and function and blocks cellular metabolic processes, thereby inducing cytotoxicity [109,120,121]. As a result of ZEA treatment, using 2,7-dichlorofluorescein diacetate, an increase in the amount of total ROS was detected in Vero [122], HepG2 [110,117,123] and HL-60 [124] cell lines. In another study, exposure of Vero cells to the ZEA metabolites α-zearalenol and β-zearalenol for 24 h also caused altered MDA levels, inhibited DNA and protein synthesis and increased levels of the oxidative stress inducible heat shock proteins, Hsp27 and Hsp70 [125]. Heat shock proteins play a role of central importance in protecting cells from damage by inflammation, oxidative stress, and other causes (Figure 3) [126].

To counteract the harmful effects of oxidative stress, cells mobilize different antioxidants, and activate damage removal and repair systems and adaptive responses. Time- and concentration-dependent glutathione (GSH) depletion was detected in the HepG2 cell line treated with ZEA [120]. However, it has not been proven that the decrease in the amount of GSH is the result of an upset oxidoreduction balance, or that the increased amount of ROS initiates GSH-consuming processes to compensate for it. In the case of Vero cells treated with ZEA, the amount of total ROS increased by 80 times leading to a 3.5-fold increment in the activity of catalase (CAT) [122].

In the study by Mike et al. [127], *Schizosaccharomyces pombe* cells not expressing β-estradiol were treated with ZEA, and the integrity of DNA, the concentration of ROS and the levels of intracellular antioxidants were observed. It was found that 500 μM ZEA treatment for 60 min induced significantly increased superoxide anione and hydrogen peroxide levels. In parallel with this, the concentration of the main antioxidant GSH was decreased, the concentrations of superoxide dismutase (SOD), CAT, glutathione reductase (GR) and glutathione S-transferase (GST) were increased, and the levels of glutathione peroxidase (GPx) and glucose-6-phosphate dehydrogenase (G6PD) were also decreased, indicating an altered oxidoreduction state of cells after ZEA exposure. DNA fragmentation and cell cycle arrest were also detected in this study. Similar results were published on swine small intestine IPEC-J2 cells [128].

The non-estrogen-specific mechanism of action published in these three groups is likely to appear also in model organisms expressing β-estradiol. Pregnant Sprague-Dawley rats were fed with 0–146 mg/kg ZEA from gestation day one to seven and the oxidoreduction parameters of jejunal tissue were measured [129]. Elevated MDA and GPx levels were noticed, while SOD and CAT activities were decreased; the change of parameters showed significant concentration-dependence. In another study, the amount of GSH, CAT and SOD levels in the testis of the examined Balb/c mice decreased as a result of chronic ZEA treatment [130]. In contrast with these findings, Boeira et al. [131] measured a significant increase in the activity of the same enzymes from the testes of mice. In bovine mammary epithelial cells, increased ROS production and apoptosis rate were observed after 24 h or 48 h of ZEA treatments at a concentration range of 0–100 μM. As biomarkers of apoptotic processes, the activity of caspase-3 was increased, the relative mRNA level of BAX was also increased and the mRNA level of the anti-apoptotic Bcl2 was decreased. The fall in mitochondrial membrane potential was also observed as another marker of oxidative stress-induced apoptosis (Figure 3, Table 2) [132].

As seen above, ZEA induces oxidative stress and apoptosis in many cell lines and this ability may play an important role in the non-estrogen-specific, oxidative stress-mediated reproductive toxicity of ZEA. In vitro chronic treatment of swine sperm cells with ZEA resulted in altered DNA fragmentation index of cells [34]. In boar semen, similar results were published. In addition to the oxidative stress-inducing and DNA integrity-affecting effects of ZEA, numerous publications have been published in which the effects on sperm function were investigated. In vitro treatment of semen samples caused a significant decline in all investigated parameters, namely viability, DNA integrity, motility and morphology [47]. In an in vivo experiment by Kim et al. [133], a single i.p. dose of 5 mg/kg of ZEA induced germ cell apoptosis after 12 h of administration in male Sprague-Dawley rats. In vitro results of Lee et al. [118] also report apoptosis and autophagy of the investigated spermatogonia cell line. Pre-exposure of bovine endometrial epithelial cells to ZEA induced pro-inflammatory cytokines (TNFA, IL1B), and chemokine IL8 mRNA upregulation resulted in altered sperm–uterine crosstalk which was manifested in suppressed sperm motility [134]. ZEA is capable of negatively influencing spermatogenesis by its oxidative stress-inducing ability. Effects of ZEA were described on several elements of spermatogenesis (e.g., Sertoli cells, Leydig cells and blood-testis barrier). Multiple factors are needed for successful spermatogenesis, but Sertoli cells and Leydig cells are crucial. During the different stages of spermatogenesis, spermatogenic cells are in close contact with Sertoli cells. Sertoli cells extend from the basement membrane to the lumen of the seminiferous tubule and provide structural and metabolic support for developing germ cells. Secretums of Sertoli cells drive the: (i) initiation of meiosis, (ii) protection of germ cells from the immune system via the blood-testis barrier, (iii) deterioration of Müllerian duct, and (iv) concentration of testosterone by secreting androgen-binding protein. In the production of androgens/testosterone, Leydig cells play a key role [135]. The deterioration of these cells and structures leads to decreased fertility. In Sertoli cells, in vitro exposure to ZEA also caused ROS induction and apoptosis [36,45]. Moreover, in the study by Zheng et al. [92], ZEA impaired Sertoli cell division and cytoskeletal structure via endoplasmic reticulum stress–autophagy–oxidative processes; the hypothesis of ZEA-induced oxidative stress was further supported by the fact that the ROS production and the degeneration of the cytoskeletal structure were alleviated by the combined treatment with N-acetyl-cysteine. Chronic, 7-day-long feeding of Balb/c mice with 40 mg/kg ZEA showed ultrastructural changes of testis such as mitochondrial swelling and degeneration, and destroyed the blood-testis barrier [46]. In Leydig cells, the same effects of ZEA have been recorded, similar to the case of Sertoli cells. In a murine Leydig tumor cell line, 0–50 μg/mL ZEA treatment for 24 h resulted in a concentration-dependent decrease in cell viability; in parallel with this, the ratio of apoptotic cells and the activity of its markers increased. The process of apoptosis took place via the endoplasmic reticulum stress-dependent signaling pathway. The apoptosis of Leydig cells leads to altered testosterone synthesis [37,42,48]. It was observed in Sertoli cells that in vitro ZEA treatment not only activates the mitochondria-dependent intrinsic pathway of apoptosis, but ZEA is also able to initiate programmed cell death via the Fas–Fas ligand extrinsic pathway. In addition, co-exposure with the caspase-8 inhibitor Z-IETD-FMK resulted in reduced decrement in mitochondrial membrane potential and apoptosis rate of ZEA-treated Sertoli cells (Figure 2 and Figure 3) [136].

Independently of ZEA, many papers have been published about the connection of oxidative stress and male infertility (for reviews, see [137,138,139,140,141]). The altered oxidoreduction state of sperm cells may contribute to a reduced probability of pregnancy. For example, blastocysts grown from oocytes fertilized with sperm that had been exposed to xanthine-xanthine oxidase displayed decreased blastocyst developmental competence by increasing blastomere DNA fragmentation [142]. The reaction of xanthine-xanthine oxidase generates O_2_^.−^ and H_2_O_2_ [143].

**Table 2 ijms-24-01578-t002:** Main data characteristic of reviewed studies which deal with non-estrogen-dependent effects.

First Author	Reference Number	Species	In Vitro/In Vivo	Method (Oral, ip, etc.)	Dose	Exposure Period	Results
Benzoni	[34]	Swine sperm	In vitro	Single exposure	10^−8^–20 μM ZEA, α-zearalenol and β-zearalenol	24 h and 48 h	Reduced cell viability and motility, increased rate of apoptosis and DNA fragmentation.
Bielas	[35]	Wild boars	In vivo	(i) Orally once a day for 7 days in every 2 months, (ii) orally once a day	(i) 150 μg/kg bw and (ii) 50 μg/kg bw ZEA	1 y	Reduced sperm movement parameters.
Chao	[36]	Sertoli cells	In vitro	Single exposure	0–50 μM ZEA	24 h and 48 h	Reduced cell viability, increased rate of apoptosis and ROS production, increased expression of γH2AX and RAD51 DNA repairing enzymes, decrement in the expression of occludin and connexin 43, proteins that are present in the testis–blood barrier and gap junctions of Sertoli cells, respectively.
Chen	[37]	Mouse Leydig (TM3) cells	In vitro	Single exposure	50 μM/L ZEA	24 h	Increased rate of apoptosis via PTEN, inhibition of PI3K/AKT signal pathway.
Filannino	[38]	Stallion sperm	In vitro	Single exposure	1 pM–0.1 mM ZEA, α-zearalenol and β-zearalenol	2 h	Hyperactivated motility of equine sperm cells, premature completion of the acrosome reaction and diminished sperm physiology. The α form of zearalenol still possessed the estrogenic ability to induce hyperactivated motility, whereas its β stereo-isomere had lost this property.
Krejcárková	[40]	Pig sperm	In vitro	Single exposure	0–20 μM ZEA and α-zearalenol	2 h and 4 h	Inhibitory effects of ZEA concentrations above 5 μM on sperm motility.
Savard	[45]	Immature Sertoli TM4 cell line	In vitro	Single exposure	0–100 μM ZEA	24 h	Activation of MAPK signaling pathway, increased ROS formation, ZEA could be detrimental to the early steps of Sertoli cell differentiation.
Tassis	[47]	Boar semen	In vitro	Single exposure	62.8 μM ZEA	4 h	Reduced cell viability and increased rate of cellular abnormalities.
Wang	[48]	Rat Leydig cells	In vitro	Single exposure	0–20 μg/mL ZEA	12 h	Inhibition of cell proliferation and increased rate of apoptosis, increased Bax expression and cytochrome c release, activation of Caspase-3 and Caspase-9, cleavage of PARP, upregulation of LC3-II and Beclin-1.
Tsakmakidis	[89]	Pietrain boar semen	In vitro	Single exposure	0–250 μm ZEA	4 h	Decreased sperm motility and viability, disrupted acrosome reaction.
Zheng	[92]	Rat Leydig cells	In vitro	Single exposure	0–20 μg/mL ZEA	24 h	Disruption of α-tubulin filaments and F-actin bundles, and damage to the nucleus of Sertoli cells. Decrease in the levels of inhibin-β and transferrin in the cultural supernatants.
Xu	[93]	Sprague Dawley rats Sertoli cells	In vitro	Single exposure	0–100 μM ZEA	24 h	Increased ratio of Bax/Bcl-2and expressions of FasL, caspases-3, 28, and 29. Reduced proliferation of Sertoli cells, induction of apoptosis and necrosis in cells via extrinsic and intrinsic apoptotic pathways.
Abid-Essefi	[109]	Vero and Caco-2 cells	In vitro	Single exposure	0–40 μM ZEA	24 h	Reduces cell viability correlated to cell cycle perturbation, inhibited protein and DNA syntheses and increased MDA formation.
Gazzah	[110]	HepG2 cells	In vitro	Single exposure	0–100 µM ZEA	60 h	Induction of Hsp70 protein, increased ROS formation, DNA fragmentation and cell-cycle arrest. Increased Bax expression, decreased Bcl-2 expression and mitochondrial membrane potential, increase cytochrome c release, activation of caspase-3 and caspase-9.
Kouadio	[112]	Caco-2 cells	In vitro	Single exposure	0–150 μM ZEA, deoxynivalenol and fumonisin B1	72 h	The three mycotoxins inducing lipid peroxidation (MDA production) in potency order fumonisin B1 > deoxynivalenol > ZEA. This effect seems to be related to their common target that is the mitochondria. Deoxynivalenol and ZEA also adversely affect lysosomes in contrast to fumonisin B1. The three mycotoxins inhibit protein synthesis. DNA synthesis seems to be restored for fumonisin B1 and deoxynivalenol suggesting a promoter activity.
Ouanes	[114]	Vero cells and Balb/c male mice	In vitro and in vivo	Single exposure	0–20 μM and 0–40 mg 6 kg bw ZEA	24 h and 48 h	ZEA induce micronuclei in cultured Vero cells as well as in mouse bone marrow cells. Vitamin E was found to prevent these toxic effects, most likely acting either as a structural analogue of ZEA or as an antioxidant.
Bouaziz	[116]	HepG2 cells	In vitro	Single exposure	0–120 µM ZEA, 0–60 nM T-2 toxin, 0–100 µM ochratoxin	0–48 h	The three mycotoxins-induced a caspase-dependent mitochondrial apoptotic pathway. The mitochondrial alterations include: bax relocalisation into the mitochondrial outer membrane, loss of the mitochondrial transmembrane potential, PTPC opening, and cytochrome c release. In the presence of ZEA and T-2 toxin, ROS level was highly increased at an early stage even before mitochondrial alterations were observed.
Ayed-Boussema	[117]	HepG2 cells	In vitro	Single exposure	0–220 μM ZEA	24 h	Reduces cell proliferation, induced an upregulation of ATM and p53 genes family accompanied by an upregulation of GADD45 to arrest the cell cycle and to allow the repair mechanisms to take place. Increased the ratio of pro-apoptotic factors/anti-apoptotic factors which led to the loss of mitochondrial potential, Bax translocation and cytochrome c release.
Lee	[118]	GC-1 spermatogonia cells	In vitro	Single exposure	0–100 μM ZEA	24 h	The cleaved caspase-3 and -8, Bad, Bax, and phosphorylation of p53 and ERK1/2, were increased. Expression levels of the autophagy-related genes Atg5, Atg3, Beclin 1, LC3, Ulk1, Bnip 3, and p62 were higher. The protein levels of both LC3A/B and Atg12 were remarkably increased. ZEA has toxic effects on tGC-1 spermatogonia cells and induces both apoptosis and autophagy.
Hassen	[120]	HepG2 cells	In vitro	Single exposure	0–350 μM ZEA	3 h and 24 h	Reduced cell viability, increased rate of DNA breaks, elevated Hsp70 and 90, reduced total GSH in cells.
Abid-Essefi	[121]	Caco-2 cells	In vitro	Single exposure	0–100 μM ZEA, α-zearalenol and β-zearalenol	48 h	Reduced cell viability, increased MDA level, increased DNA fragmentation, increased Caspase-3 activity and decreased Bcl-2 level.
Abid-Essefi	[122]	Vero cells	In vitro	Single exposure	0–120 μM ZEA	24 h	Reduced cell viability, increased ROS production, increased CAT activity and DNA damage. Reduced negative effects after co-treatment of an antioxidant.
Bennour	[123]	HepG2 and Vero cells	In vitro	Single exposure	0–120 μM ZEA	0–40 h	Inhibited cell proliferation, induction of Hsp70 and 27, increased ROS production, antioxidants protect against ZEA.
Stadnik	[124]	Liver cells of male Wistar rats	In vivo	Orally once a day	50–500 μg/kg bw ZEA	10 d	Reduced activities of SOD and GPx.
Othmen	[125]	Vero cells	In vitro	Single exposure	0–180 μM α-zearalenol and β-zearalenol	24 h	Increased MDA level, inhibited protein and DNA synthesis, elevated Hsp27 and 70.
Mike	[127]	*Schizosaccharomyces pombe*	In vitro	Single exposure	0–500 μM ZEA	1 h	Decreased activities of GSH, GPx and G6PD, increased activities of SOD, CAT, GR and GST, increased ROS production, DNA fragmentation and cell cycle arrest.
Fan	[128]	IPEC-J2	In vitro	Single exposure	0–8 μg/mL ZEA	2 h and 48 h	Increased MDA level and ROS production, decreased GSH, CAT and SOD activities, decreased mitochondrial membrane potential.
Liu	[129]	Jejunal tissue of pregnant Sprague-Dawley rats	In vivo	Orally once a day	0–146 mg/kg bw ZEA	Gestation day 1 to 7	Increased MDA and GPx levels, decreased SOD and CAT activities.
Salah-Abbés	[130]	Balb/c mice	In vivo	Orally once a day	40 mg/kg bw ZEA	28 d	Increased MDA level, decreased GPx, CAT and SOD activities, reduced testis, seminal vesicle and prostate weights, diminished sperm count and motility.
Boeira	[131]	Liver, kidney and testis of Swiss albino mice	In vivo	Single exposure, orally	40 mg/kg bw ZEA	48 h	Increased CAT and SOD activities, reduced GST activity.
Fu	[132]	Bovine mammary epithelial cell line MAC-T	In vitro	Single exposure	0–100 μM ZEA	24 h and 48 h	Reduced cell viability, increased ROS production, reduced mitochondrial membrane potential, altered expression of endoplasmic reticulum stress-related genes (Grp78, Hsp70, Eif2a, Atf6, Ask1 and Chop).
Kim	[133]	Male Sprague-Dawley rats	In vivo	Single exposure, i.p.	5 mg/kg bw ZEA	0–48 h	Germ cell degeneration and apoptosis.
Elweza	[134]	Bovine endometrial epithelial cells	In vitro	Single exposure	0–1000 ng/mL ZEA	24 h	Increased expression of inflammatory cytokines TNFA, IL1B and chemokine IL8, altered sperm-uterine crosstalk, reduced sperm motility.
Cai	[136]	Sertoli cells	In vitro	Single exposure	0–20 μM/L	24 h	Increased rate of apoptotic cells, decreased expression of Bcl-2, increased expression of Bax, tBID, Fas, FasL, FADD, Caspase-8 and Caspase-9, increased release of cytochrome c.

bw: body weight; s.c.: subcutaneous injection; i.p.: intraperitoneal injection.

## 5. Summary

ZEA, like many other mycotoxins, has a great impact on animals’ and humans’ health and several papers have been published about its effects on female and male reproduction. This last section summarizes a working model characterizing estrogen-dependent and non-estrogen-dependent modes of action by which ZEA has an adverse impact on the quality of male reproductive cells to impede fertility. Since ZEA is analogous to the hormone 17β-estradiol, it is capable competitively binding to the receptors of estrogen, influencing the hormone homeostasis. The increment in the estradiol level induces an inhibitory effect on the pituitary gland, LH level decreases and testosterone synthesis of Leydig cells also decreases. Altered testosterone leads to diminished spermatogenesis. ERs can be generally found on sperm cells, Leydig cells and Sertoli cells, therefore ZEA has an adverse effect on the whole male reproductive system via epigenetic pathways; not only spermatogenesis is affected, but parameters of sperm cells will also negatively change (e.g., viability, motility, DNA integrity, acrosome reaction, hyperactivity, etc.).

The oxidative stress-inducing ability of ZEA has been investigated in several ER non-expressing model organisms (similar results were published in ER-expressing models as well). ZEA is an inducer of ROS, whose main sources are the endoplasmic reticulum and the mitochondria. ROS can damage every cellular component such as unsaturated fatty acids of lipids, the cytoskeletal system and the DNA itself. In response to an increased ROS level, the synthesis of many antioxidant molecules is activated. If the ROS production exceeds the synthesis of antioxidants, programmed cell death may occur. ZEA induces apoptosis via both intrinsic and extrinsic apoptotic pathways leading to cell death. Altered caspase 3, caspase 8, Bax, Bad Erk1/2, Bcl2 expressions and mitochondrial membrane potential can be found in the literature.

As we have seen previously, ZEA has a significant negative effect on male reproduction in both estrogenic and non-estrogenic pathways. However, based on the data of the literature, it is unfortunately not possible to decide in all cases whether the effect of the toxin is due to the estrogenic effect or the non-estrogenic effect. In many cases, estrogen-dependent and non-estrogen-dependent effects are the same. Another major shortcoming of the literature is that the number of in vivo experiments is very low, and even in the available data, there is a large variance in the method of treatment, exposure duration and applied dose. Most papers investigate only ZEA, and there is only a limited number of studies on ZEA derivatives and conjugates. However, since these compounds exert their essential effect soon after exposure, in vitro studies and methods are not enough to represent the changes and mode of actions that might be observed when whole organism is exposed. It follows that the number of articles dealing with estrogen-specific effects of ZEA is much lower than those dealing with its oxidative stress-inducing effects (Table 1 and Table 2). Further investigations are required.

## Figures and Tables

**Figure 1 ijms-24-01578-f001:**
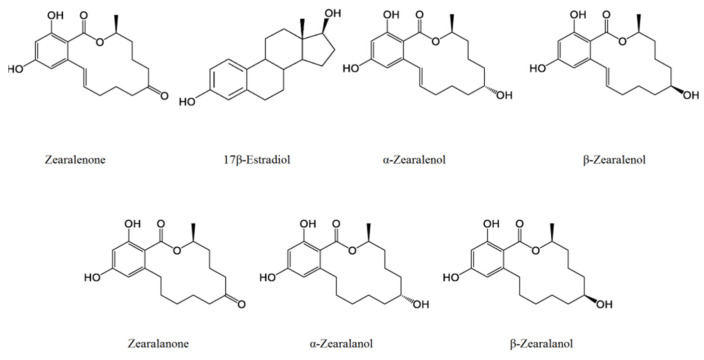
Chemical structure of zearalenone and its main metabolites and their structural similarity to 17β-estradiol.

**Figure 2 ijms-24-01578-f002:**
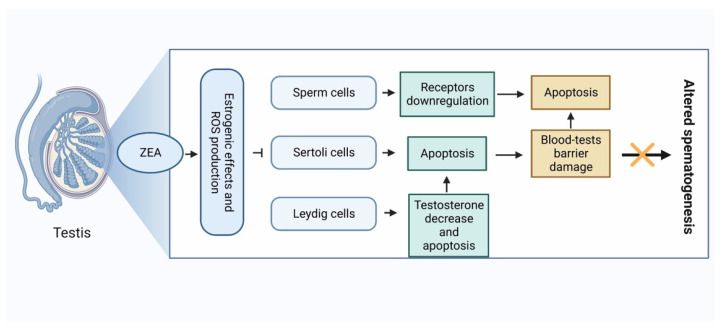
Mechanism of action of zearalenone on male reproductive cells.

**Figure 3 ijms-24-01578-f003:**
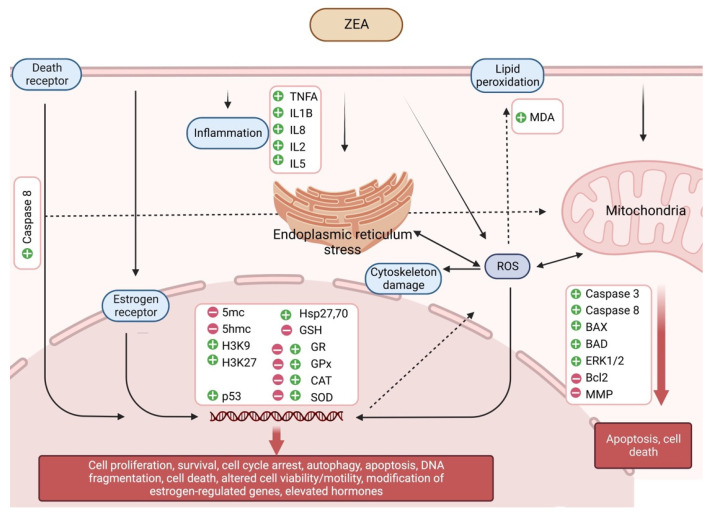
Estrogenic and non-estrogenic effects of zearalenone.

**Table 1 ijms-24-01578-t001:** Main data characteristic of reviewed studies which deal with estrogen-dependent effects.

First Author	Reference Number	Species	In Vitro/In Vivo	Method (Oral, ip, etc)	Dose	Exposure Period	Results
Gao	[39]	CD-1 male mice	In vivo	Orally once a day	20 μg/kg and 40 μg/kg bw ZEA	5 w	Disrupted process of meiosis, inhibited spermatogenesis and diminished semen quality with the decrease in spermatozoa motility and concentration. The DNA methylation markers 5mC and 5hmC were decreased, the histone methylation marker H3K27 was increased, at the same time estrogen receptor alpha was diminished.
Lin	[42]	MLTC-1 cells	In vitro	Single exposure	0–200 μg/mL ZEA	24 h	Reduced cell viability, increased rate of apoptosis, increased caspase-3 activity, reduced testosterone levels.
Liu	[43]	Mouse Leydig cells	In vitro	Single exposure	0–20 μg/mL	24 h	Reduced production of testosterone via crosstalk of estrogen receptor signaling, elevated cellular cAMP levels, reduced mitochondrial membrane potential, decreased expression of P450scc, 17b-HSD, and P450c17, increased expression of StAR and 3b- HSD, Nur77 expression was significantly inhibited.
Long	[44]	Kunming male mice	In vivo	Once a day, i.p.	0- 75 mg/kg bw ZEA	5 d	Reduced weights of testis and epididymis, sperm tail defects and head abnormalities, reduced concentration of testosterone, increased enzyme activities of LDH, AKP and ACP, increased mRNA expressions of Vim and Cldn11.
She	[46]	Male Balb/c mice and TM4 mouse Sertoli cells line	In vivo and in vitro	Orally once a day and single exposure	40 mg/kg bw and 0–20 μM/L ZEA	5–7 d and 24 h	Ultrastructural changes in testis, reduced sperm motility, altered expression of blood-testis-barrier proteins and autophagy-related proteins, increased level of cytoplasmic Ca^2+^.
Filipiak	[49]	Wistar rats	In vivo	Once a day, s.c.	4 or 40 μg ZEA	10 d	Reduced testes weight, seminiferous tubule diameter and length, reduced number of Sertoli cells.
Zhou	[50]	Sprague Dawley rats and immature Leydig cells	In vivo and in vitro	Once a day, intratesticularly and single exposure	0–300 ng/testis and 0–50 μM ZEA	21 d and 24 h	Reduced serum testosterone levels, reduced Leydig cell number and cell-specific gene/protein expression, reduced expression of steroidogenic factor 1 (Nr5a1), inhibited androgen production and steroidogenic enzyme activities in immature Leydig cells by downregulating expression levels of cholesterol side cleavage enzyme (Cyp11a1), 3β-hydroxysteroid dehydrogenase 1 (Hsd3b1), and steroid 5α-reductase 1 (Srd5a1).
Men	[81]	ICR pregnant mice and its male offspring	In vivo	Orally once a day	0–40 µg/kg bw ZEA	7 d	Disrupted meiosis, alteration in DNA/histone methylation (reduced 5hmC, increased H3K27me3, H3K9me2, G9a), higher ratio of ERα-positive Leydig cells, decreased sperm motility and concentration, disrupted sperm acrosome integrity, decreased body-weight, reduced testis weight, liver damage - increased AST.
Pang	[90]	CD-1 male mice	In vivo	Orally once a day	20 or 40 μg/kg ZEA	42 d	The spermatogenic cells were declined, increased rate of DNA breaks.
Koraïchi	[91]	Sprague Dawley rats and SerW3 Sertoli cell line	In vivo and in vitro	Once a day s.c. and single exposure	0–100 μg /day ZEA	5 d post-natal	Modulations of mRNA and protein levels of Abcb1, Abcc1, Abcg2, Abcc4 and Abcc5 were observed, along with Abcc4 protein cellular delocalization.
Frizell	[105]	RGA cell lines were generated from human mammary gland cell and H295R human adrenocortical carcinoma cells	In vitro	Single exposure	0–100 μM ZEA, α-zearalenol and β-zearalenol	48 h	α -zearalenol exhibited the strongest estrogenic potency, slightly less potent than 17 β - estradiol. ZEA was ∼70 times less potent than α -zearalenol and twice as potent as β -zearalenol. Binding of progesterone to the progestagen receptor was shown to be synergistically increased in the presence of ZEA, α -zearalenol or β -zearalenol. ZEA, α -zearalenol or β -zearalenol increased production of progesterone, estradiol, testosterone and cortisol hormones in the H295R steroidogenesis assay.

bw: body weight; s.c.: subcutaneous injection; i.p.: intraperitoneal injection.

## Data Availability

Not applicable.

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
