# Peer review of "Estrogenic and Non-Estrogenic Disruptor Effect of Zearalenone on Male Reproduction: A Review"

_ijms, 2023, doi:10.3390/ijms24021578_

Round 1

Reviewer 1 Report

Review paper by AndrásBalló et al aimed to present estrogenic and non-estrogenic disruptor effect of zearalenone on male reproduction. Paper is well written and topic is important. However, the manuscript would benefit from further corrections and clarification before being acceptable for publication.

In the abstract, it is necessary to provide brief information about the other adverse effects that might be caused by ZEA (not only reproductive effects). (i) and (ii) should be removed from the abstract since there is no clear listing in the text.

The authors should provide tables for each of the sections discussing the main ZEA effects (estrogenic effects/Non-estrogenic). In these tables, all of the mentioned studies should be listed, including the most important findings, but also valuable information, such as: type of the study (in vitro/in vivo), type of cells/animals used, applied concentrations/doses, duration of the exposure, exposure route, etc.

Additionally, in vitro studies should be clearly separated from in vivo studies, while I would even suggest adding more in vivo studies, since in vitro methods are not enough to represent changes that might be observed when whole organism is exposed.

The authors should provide criteria of the selection for the studies included in the manuscript.

Author Response

Answers to Review 1

Thank you for your rapid response and constructive criticism. Most of suggestions have been accepted and corrected.

Suggestion:

In the abstract, it is necessary to provide brief information about the other adverse effects that might be caused by ZEA (not only reproductive effects). (i) and (ii) should be removed from the abstract since there is no clear listing in the text. 

Answer:

Abstract has been modified based on the suggestions.

Suggestion:

The authors should provide tables for each of the sections discussing the main ZEA effects (estrogenic effects/Non-estrogenic). In these tables, all of the mentioned studies should be listed, including the most important findings, but also valuable information, such as: type of the study (in vitro/in vivo), type of cells/animals used, applied concentrations/doses, duration of the exposure, exposure route, etc. 

Answer:

The requested tables have been made and attached into the manuscript.

Suggestion:

Additionally, in vitro studies should be clearly separated from in vivo studies, while I would even suggest adding more in vivo studies, since in vitro methods are not enough to represent changes that might be observed when whole organism is exposed. 

Answer:

Unfortunately, due to the shortness of the available time, we did not have the opportunity to include additional in vivo studies. However, in the "Summary" section, we explain your suggestion that only this type of study is suitable for examining the full and actual mechanism of action.

Suggestion:

The authors should provide criteria of the selection for the studies included in the manuscript. 

Answer:

The following paragraph has been added to the manuscript:

“Studies that met all the following criteria were included: (i) investigated the impact of ZEA on the reproductive system of male model organisms (rodents, pigs, etc.); (ii) investigated the effects of ZEA on male germ cells like spermatozoa, Sertoli cells and Leydig cells. (iii) Studies conforming to any of the following criteria were also included: investigated the oxidative stress and apoptosis-inducing ability of ZEA on estrogen receptor expressing and non-expressing cells and cell lines (spermatozoa, yeasts, HepG2, HL-60, Vero, Caco-2, etc.)”

Sincerely,

Gábor Máté

Reviewer 2 Report

This manuscript discusses estrogenic and non-estrogenic disruptor effect of zearalenone on male reproduction, and two types of mode of action of ZEA have been described. In fact it also gives an overview of ZEA's immunotoxicity, genotoxicity, carcinogenicity, reproductive toxicity and endocrine disruption effects.  I suggest the manuscript could be accepted for publication after minor revisions. 

Some ZEA derivatives such as α-zearalenol are more toxic than ZEA. ZEA derivatives can also be reconverted to ZEA in humans and animals, leading to the accumulation of toxins. I believe that the discussion of the conversion between ZEA and its derivatives in this paper is inadequate.

There are two titles both with "3."(3. Chemical properties. 3.Overview of the pathogenesis of zearalenone). Please check.

This chapter "3. Chemical properties" seems to be of little significance, and I am more concerned with the parameters such as, melting point, boiling point, XLogP or LogKow...

Author Response

Answers to Review 2

Thank you for your rapid response and positive opinion. All of suggestions have been accepted and corrected.

Suggestion:

Some ZEA derivatives such as α-zearalenol are more toxic than ZEA. ZEA derivatives can also be reconverted to ZEA in humans and animals, leading to the accumulation of toxins. I believe that the discussion of the conversion between ZEA and its derivatives in this paper is inadequate.

Answer:

The manuscript was supplemented with some sentences about ZEA metabolites. The reconversion to ZEA was avoided because most of available data come from in vitro studies with genetically modified microorganisms. In human and animal models, the authors did not find suitable publication to support this reconversion in vivo.

Suggestion:

There are two titles both with "3."(3. Chemical properties. 3.Overview of the pathogenesis of zearalenone). Please check.

Answer:

Thanks for the observation, the titles have been corrected.

Suggestion:

This chapter "3. Chemical properties" seems to be of little significance, and I am more concerned with the parameters such as, melting point, boiling point, XLogP or LogKow...

Answer:

As requested, additional chemical parameters have been added.

Sincerely,

Gábor Máté

Round 2

Reviewer 1 Report

Changes and corrections made to improve the manuscript were appropriate.